# Filling the Upper Pole with the Pectoralis Major Muscle Flap in Profunda Femoris Artery Perforator Flap Breast Reconstruction

**DOI:** 10.3390/medicina58040458

**Published:** 2022-03-22

**Authors:** Hidehiko Yoshimatsu, Hiroki Miyashita, Ryo Karakawa, Yuma Fuse, Tomoyuki Yano

**Affiliations:** Department of Plastic and Reconstructive Surgery, Cancer Institute Hospital, Japanese Foundation for Cancer Research, Tokyo 135-8550, Japan; hirokimiya@gmail.com (H.M.); ryo.kyara@gmail.com (R.K.); yuyuma.fuse@gmail.com (Y.F.); yanoaprs@icloud.com (T.Y.)

**Keywords:** profunda femoris artery perforator flap, pectoralis muscle flap, autologous breast reconstruction

## Abstract

*Background and Objectives*: Among many donor site options for autologous breast reconstruction, the use of the profunda femoris artery perforator (PAP) flap has become common in patients who are not suitable for the gold standard procedure, the deep inferior epigastric artery perforator flap. However, its limited volume has precluded its wide use in breast reconstruction. The aim of this report was to demonstrate the effectiveness of a method in which the anatomical position of the pectoralis major muscle was adjusted to augment the volume of the superior pole of the breast during PAP flap transfer. A comparison was made with a conventional PAP flap breast reconstruction. *Materials and Methods*: Fifty-nine consecutive cases where unilateral autologous breast reconstruction was performed using the vertically designed PAP flap were retrospectively reviewed. Conventional PAP flap transfer was performed in 36 patients (Group 1), and PAP flap transfer with pectoralis major muscle augmentation was performed in 23 patients (Group 2). *Results*: The patient satisfaction at 12 months postoperatively was statistically greater in Group 2, with the pectoralis major muscle augmentation, than in Group 1 [23/36 (64%) vs. 22/23 (96%), *p* = 0.005]. There were no significant differences in postoperative complication rates at the reconstructed site [2/36 (5.6%) vs. 0/23 (0%), *p* = 0.52]. *Conclusions*: Higher patient satisfaction could be achieved with pectoralis major muscle augmentation in PAP flap breast reconstruction without increasing the postoperative complication rate at the reconstructed site.

## 1. Introduction

Autologous breast reconstruction using free flaps has become a standard procedure for patients who have received radiation therapy, refuse reconstruction using silicone implants, or are likely to undergo extensive skin excision. The deep inferior epigastric artery perforator (DIEP) flap is the gold standard for autologous breast reconstruction, but other donor sites, including the thigh region and the buttock, have been used for patients with less abdominal fat, patients with previous abdominal surgery, and patients who refuse the abdomen as the flap donor site [1,2,3,4,5,6,7,8,9]. Among the flaps harvested from these donor sites, the vertically-designed profunda femoris artery perforator (PAP) flap was proposed to mitigate the following problems entailing the transverse upper gracilis (TUG) flap and the transversely designed PAP flap: less flap volume, scar migration, and labial spreading [5,6]. However, even with the vertically designed PAP flap, insufficient volume still remains the primary concern, often resulting in a lack of volume in the upper pole of the breast.

In this article, we aimed to examine whether a method in which the anatomical position of the pectoralis major muscle is adjusted to add volume to the superior pole of the breast during PAP flap transfer is effective in increasing patient satisfaction by comparing the results after PAP flap autologous breast reconstruction with and without pectoralis major muscle flap augmentation.

## 2. Materials and Methods

### 2.1. Study Design and Population

This retrospective analysis examined all medical records of consecutive patients with unilateral breast cancer who underwent breast reconstructions using the PAP flap between April 2018 and May 2020. The study was approved by the institutional review board of the Cancer Institute Hospital of the Japanese Foundation for Cancer Research (Approval number: CT2021―0099). All patients had small to medium breast size and lacked adequate abdominal tissue or were contraindicated for the DIEP flap transfer.

Eligible patients were divided into two groups based on the reconstruction approach. In Group 1, the PAP flap was transferred without the use of the pectoralis major muscle flap. In Group 2, the transferred PAP flap was augmented with the pectoralis major muscle flap.

For assessment of patient satisfaction, the following self-assessment question was asked 12 months postoperatively after the breast reconstruction: “Considering the volume and shape of my breast, I am satisfied with my reconstruction”. The question used a 5-point Likert scale with 1 indicating high satisfaction and 5 indicating low satisfaction. Only responses of 1 or 2 were rated as “satisfied”.

### 2.2. Study Outcomes and Variables

Data collected included patient demographic information (age and body mass index) and the timing of reconstruction. Data on flap weight (g) were obtained from operative notes. Data collected during hospitalization included postoperative complications at the reconstructed site (arterial and venous thrombosis, dehiscence, and bleeding). Patient satisfaction on cosmesis of the reconstructed breast was obtained at the 12-month follow-up.

The primary outcome was patient satisfaction at the 12-month follow-up. The secondary outcome was postoperative complications at the reconstructed site.

### 2.3. Statistical Analysis

Patient characteristics were compared across the two groups using Fisher’s exact test for categorical variables: timing of reconstruction, complications, and patient satisfaction. We compared continuous variables across the two groups using the Student’s t-test for age, body mass index, and flap weight (g). All statistical tests were two-sided, and a value of *p* < 0.05 was considered statistically significant. All statistical analyses were performed using SPSS v.23.0 (IBM Corp., Armonk, NY, USA).

### 2.4. Surgical Technique of the Pectoralis Muscle Flap Augmentation

The PAP flap is elevated in the frog-leg position. Preoperatively, the perforators are marked using a handheld Doppler ultrasound device. The lymphatic collector vessels are marked using indocyanine green lymphography. The skin paddle is designed in a vertical fashion so as not to disrupt the lymphatic pathway [6,10]. The origins of the pectoralis major muscle are detached along the inframammary fold. The attachment of the muscle to the sternum is left intact. This procedure results in cephalad repositioning of the pectoralis major muscle. The muscle flap is then rolled up to add volume to the superior pole of the breast. Sutures were added to the muscle flap itself to retain its shape. The PAP flap is anastomosed to the internal mammary artery and to the internal mammary vein and fixed to reconstruct the lower pole. The caudal edge of the muscle roll is sutured to the cephalad edge of the PAP flap to reconstruct a smooth plane (Figure 1 and Figure 2). In both groups, the anastomoses were performed either in the third or the fourth intercostal space. In Group 2, the anastomoses were performed beneath the pectoralis major muscle flap.

## 3. Results

A total of 59 patients (59 PAP flaps) were selected. PAP flaps augmented with the pectoralis muscle flap were performed in 23 patients. Table 1 presents the background characteristics of patients in the two groups. No significant difference was found between the groups, with the exception of reconstruction timing. Immediate reconstruction was more frequently seen in Group 1 without use of the pectoralis major muscle flap. Recipient vessels were the internal mammary artery and vein in all cases.

Table 2 presents the results of the primary and secondary outcomes. The patient satisfaction at 12 months postoperatively was statistically greater in Group 2, with the pectoralis major muscle augmentation, than in Group 1 [23/36 (64%) vs. 22/23 (96%), *p* = 0.005]. There were no significant differences in postoperative complication rates at the reconstructed site [2/36 (5.6%) vs. 0/23 (0%), *p* = 0.52]. All flaps survived completely, but one case of venous thrombosis and one case of dehiscence were seen in Group 1.

### 3.1. Case Reports

#### 3.1.1. Case 1

A 40-year-old woman was diagnosed with ductal carcinoma of the left breast. Delayed breast reconstruction using the PAP flap was planned because the amount of abdominal tissue was insufficient. A vertical PAP flap with a skin paddle of 13 × 7.5 cm was harvested from the left medial thigh. The pedicle was anastomosed to the internal mammary vessels in an end-to-end fashion. The PAP flap was de-epithelialized and inset to fill the lower pole of the breast. A skin island was left for postoperative monitoring. The pectoralis major muscle roll was elevated and sutured to the PAP flap in the above-described fashion (Figure 2). The postoperative course was uneventful, and there was no donor site complication. At the 12 months follow-up, the volume of the upper pole of the reconstructed breast retained its volume (Figure 3).

#### 3.1.2. Case 2

A 37-year-old woman was diagnosed with ductal carcinoma of the left breast. Delayed breast reconstruction using the PAP flap was planned because the patient refused to use the abdomen as the flap donor site. A vertical PAP flap with a skin paddle of 14 × 6 cm was harvested from the left medial thigh (Figure 4).

The pedicle was anastomosed to the internal mammary vessels in an end-to-end fashion. The PAP flap was de-epithelialized and inset to fill the lower pole of the breast, and a skin island was left for postoperative monitoring. The pectoralis major muscle roll was elevated and sutured to the PAP flap (Figure 5).

The postoperative course was uneventful, and there was no donor site complication. At the 12 months follow-up, the volume of the upper pole of the reconstructed breast retained its volume, and the patient was satisfied with the reconstructed breast (Figure 6).

## 4. Discussion

With abundant volume and skin texture and color similar to that of the breast, the abdominal region has been the gold standard for women undergoing autologous breast reconstruction. However, there are cases where the DIEP does not come on the top of the list: patients with less abdominal fat, patients with previous abdominal surgery, and patients who refuse the abdomen as the flap donor site, often seen in nulliparous women.

The superior and inferior gluteal artery perforator flaps from the buttock, and the transverse and vertical upper gracilis (TUG and VUG) from the medial thigh region, have been introduced as alternatives for the DIEP flap [2,3,4,5]. However, an entire set of disadvantages accompany these flaps. The superior and inferior gluteal artery perforator flaps require a position change during the operation, and the pedicle dissection can be significantly tedious because the buttock fat is firmer and more fibrous than the fat in other regions [2,3]. The TUG and VUG are myocutaneous flaps involving the harvest of the gracilis muscle. The transverse scar after the TUG harvest can result in scar migration and labial spreading [4]. With the VUG, unreliable perfusion in the distal skin region is the most significant drawback. In addition, the harvest of the TUG and VUG can result in lymphedema at the donor site [5].

In 2012, Allen et al. first reported the use of the PAP flap for breast reconstruction [1]. The vertical design of the skin paddle was proposed by Scaglioni et al. to mitigate the disadvantages of the horizontally designed PAP flaps [6]. The vertical flap design was later reported to contribute to the less frequent occurrence of donor site lymphedema [10]. However, insufficient volume still remains the primary concern even with the vertically designed PAP flap, often resulting in a lack of volume in the upper pole of the breast, typically seen in our patient group who underwent PAP flap transfer without pectoralis muscle flap augmentation (Figure 7).

Harvesting too much volume from this region results in donor site morbidity such as wound dehiscence and seroma [11]. Stacked PAP flaps and stacked TUG flaps were proposed to add more volume [12,13,14]. However, harvesting the flaps from bilateral medial thighs and additional anastomoses add to the technical challenge and operative time. Lipofilling is an excellent procedure for touch-ups, but its cost cannot be overlooked [15,16]. Because breast cancer patients have already undergone a series of medical and surgical procedures, we looked for a simple, less time-consuming, and cost-effective method with the fewest number of interventions to achieve an optimal result.

In 2006, Tebbetts et al. reported on the use of the pectoralis major muscle flap to bolster the volume of the reconstructed breast while using silicone breast implants [17]. This concept was adopted in PAP flap breast reconstruction. The volume of the upper pole of the breast can be augmented when a PAP flap is used. This strategy widens the indication of the PAP flap breast reconstruction and precludes additional postoperative procedures such as lipoinjection. A smooth plane can be reconstructed because the muscle roll can be sutured to the cephalad edge of the PAP flap, which is not possible in breast reconstruction using silicone implants (Figure 2 and Figure 5).

In our case series, the use of the pectoralis muscle to add to the volume in the upper pole increased patient satisfaction at 12 months postoperatively with no increase in postoperative complications at the reconstructed site (Table 2).

The use of the pectoralis major muscle may raise anxiety over postoperative functional dysfunctions of the shoulder. Decreases in shoulder strength and range of motion have been reported after the use of the pectoralis major musculocutaneous flap in head and neck cancer treatment [18,19,20,21]. The pectoralis major muscle is comprised of the clavicular part, which contributes to flexion, horizontal adduction, and inward rotation of the humerus, and the sternocostal part, which is responsible for downward and forward movement of the arm and inward rotation when accompanied by adduction [21]. The pectoralis major musculocutaneous flap entails detachment of the whole sternocostal portion and a part of the clavicular portion, which thus results in shoulder dysfunction. The pectoralis muscle flap used in our series involves detachment of only the caudal portion or the sternocostal section. In a similar case, series using the pectoralis major muscle flap in 468 patients, including professional athletes, Tebbetts reported no complaints related to pectoralis function [17].

The limitation of this study is the short follow-up. In a long-term follow-up for more than 3 years, volume decrease due to muscle contracture of the pectoralis major muscle flap may be expected. Kang et al. reported that the volume of the muscle part of the latissimus dorsi muscle flap statistically significantly decreased until the third year postoperatively, and it decreased by 14.64% on average from 6 months to 3 years postoperatively [22]. Of note, there was no statistically significant decrease after 3 years. The greatest decrease in volume was seen in the first 12 months, which was 8.04%. Although we preserved the motor nerve of the muscle to mitigate the volume reduction, a long-term follow-up is necessary to evaluate the results after pectoralis muscle flap augmentation. Another limitation of augmentation with the pectoralis muscle flap is that the reconstructed upper pole will not be as soft as the region reconstructed with the PAP flap.

## 5. Conclusions

The upper pole can be augmented using the pectoralis major muscle flap in breast reconstruction using the PAP flap, which expands its indications. The patient satisfaction rate at 12 months after the reconstruction was statistically higher in the patient group with the pectoralis major muscle flap augmentation than in the patient group without the augmentation.

## Figures and Tables

**Figure 1 medicina-58-00458-f001:**
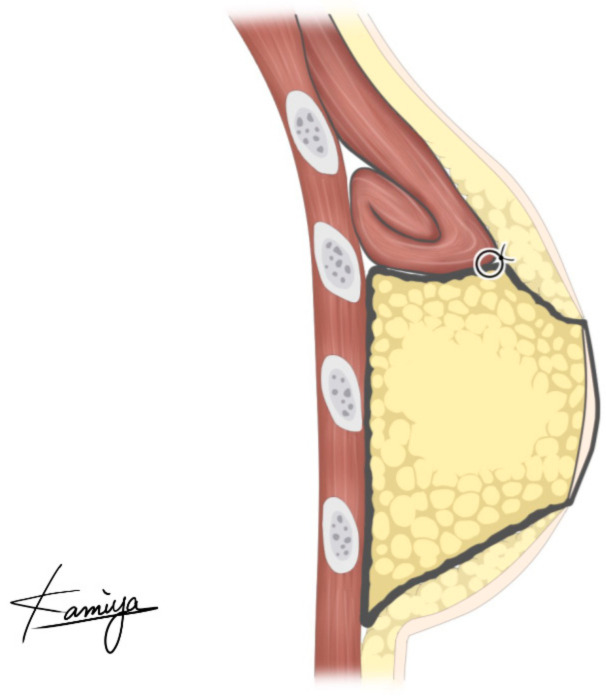
An illustration demonstrating the concept of a profunda femoris artery perforator flap breast reconstruction augmented with the pectoralis major muscle flap.

**Figure 2 medicina-58-00458-f002:**
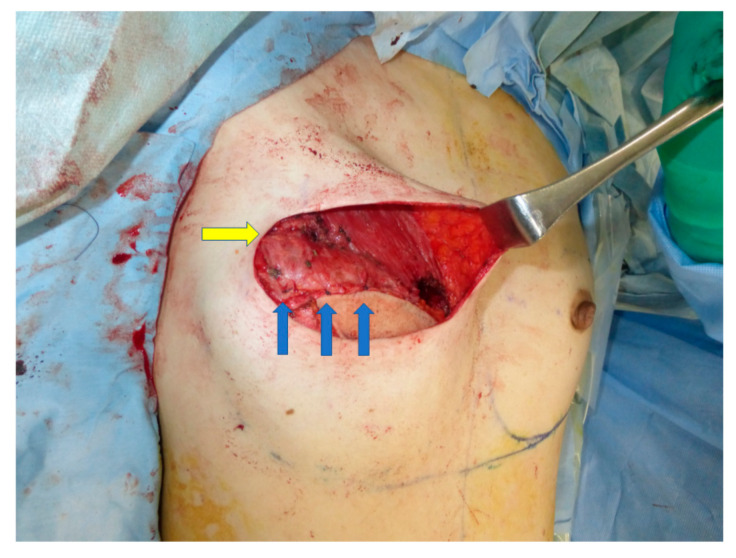
The caudal edge of the muscle roll (yellow arrow) is sutured to the cephalad edge of the PAP flap to reconstruct a smooth plane (blue arrows).

**Figure 3 medicina-58-00458-f003:**
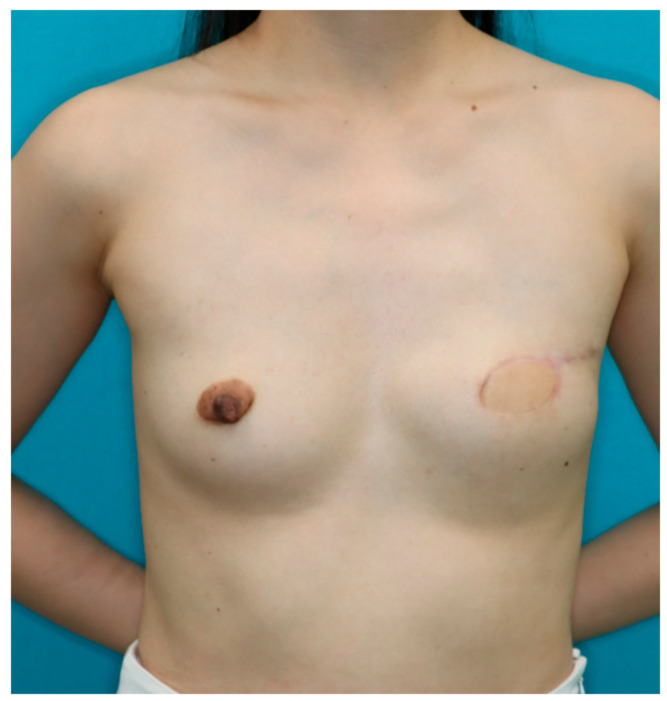
At the 12 months follow-up, the upper pole of the reconstructed breast retained its volume.

**Figure 4 medicina-58-00458-f004:**
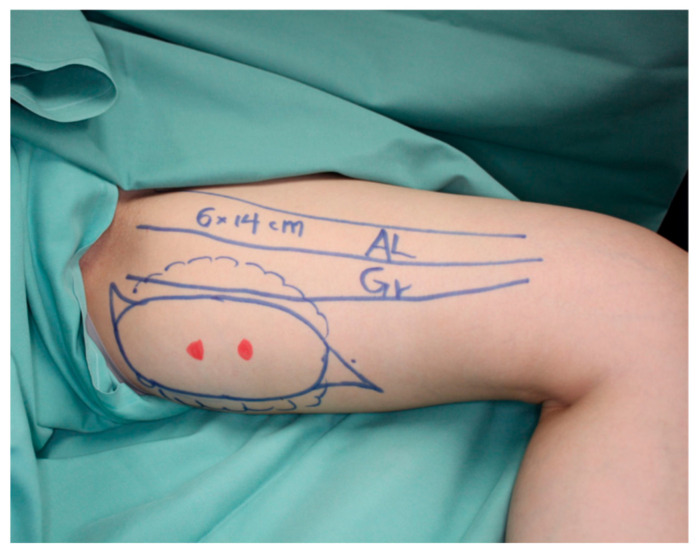
A vertical PAP flap with a skin paddle of 14 × 6 cm was harvested from the left medial thigh.

**Figure 5 medicina-58-00458-f005:**
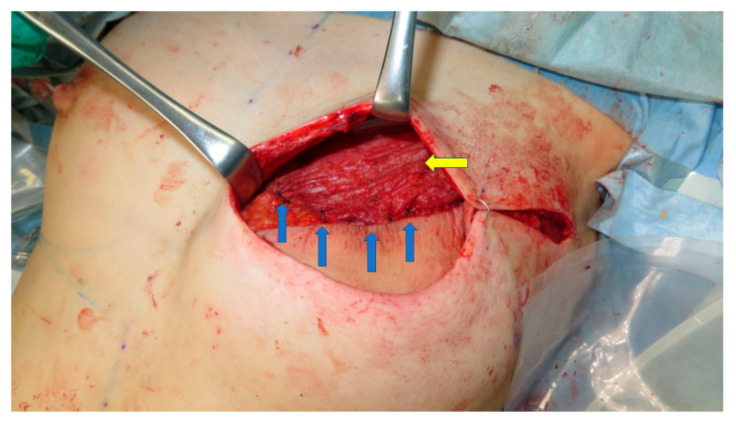
The caudal edge of the muscle roll (yellow arrow) is sutured to the cephalad edge of the PAP flap to reconstruct a smooth plane (blue arrows).

**Figure 6 medicina-58-00458-f006:**
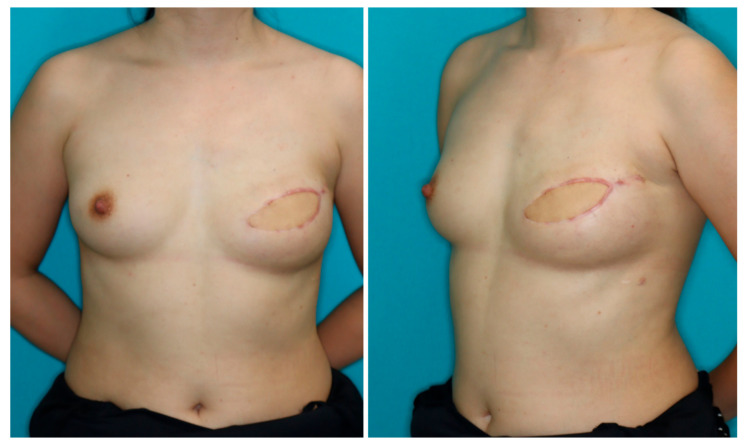
At the 12 months follow-up, the upper pole of the reconstructed breast retained its volume, and the patient was satisfied with the reconstructed breast.

**Figure 7 medicina-58-00458-f007:**
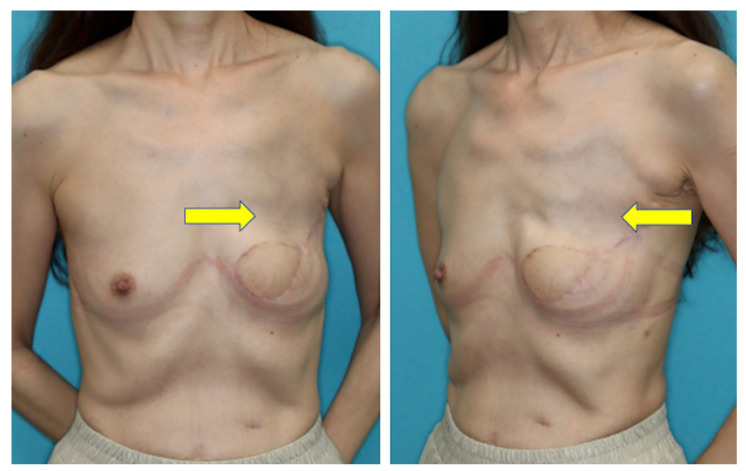
Lack of volume in the upper pole apparent at the 12 months follow-up after breast reconstruction with a conventional PAP flap (yellow arrow).

**Table 1 medicina-58-00458-t001:** Patient characteristics.

	PAP Flap without Use of the Pectoralis Major Muscle Flap Cohort (Group 1) N = 36	PAP Flap Augmented with Pectoralis Major Muscle Flap Cohort (Group 2) N = 23	*p*-Value
Age (years), mean (range)	46.6 (27–67)	47.0 (34–64)	0.91
Delayed reconstruction	19 (53%)	22 (96%)	<0.001
Body mass index, mean (range)	20.6 (17.1–29.6)	21.3 (17.0–30.0)	0.32
Flap weight (g), mean (range)	218 (105–435)	216 (140–370)	0.92

**Table 2 medicina-58-00458-t002:** Primary and secondary outcomes.

	PAP Flap without Use of the Pectoralis Major Muscle Flap Cohort (Group 1) N = 36	PAP Flap Augmented with Pectoralis Major Muscle Flap Cohort (Group 2) N = 23	*p*-Value
Satisfied after 12 months	23 (64%)	22 (96%)	0.005
Postoperative complications	2 (5.6%)	0 (0%)	0.52

## Data Availability

The data that support the findings of this study are available from the corresponding author, H.Y., upon reasonable request.

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
