# Peer review of "Filling the Upper Pole with the Pectoralis Major Muscle Flap in Profunda Femoris Artery Perforator Flap Breast Reconstruction"

_medicina, 2022, doi:10.3390/medicina58040458_

Round 1

Reviewer 1 Report

This is a paper dealing with a commonly seen medical condition after breast reconstruction.

The idea of using a rolled pectoralis major muscle flap has been previously used in breast reconstruction, as also authors mentioned, but this is the first time using it in combination with PAP flap.

The paper is well written in a clear and simple language, easy to read and interesting for breast reconstructive surgeons.

Some technical issues could be add in order to strngthen the interest of the readers:

1) In which intercostal space the authors performe the anastomosis of the flaps with the IM vessels, in both groups? In the second group do they perform the anastomosis underneath the pec major or they spread the muscle fibers?

2) When rolling the pec major do they suture the muscle to itself to keep the shape?

Please add these infromation to the paper.

Author Response

Thank you for your constructive comments concerning our manuscript “Filling the Upper Pole with the Pectoralis Major Muscle Flap in Profunda Femoris Artery Perforator Flap Breast Reconstruction.” We have studied your comments carefully and made corrections, which we hope will meet your approval. We answered your questions or comments in detail in the following paragraphs. All changes to the manuscript are written using "Track Changes."

1) In which intercostal space the authors performe the anastomosis of the flaps with the IM vessels, in both groups? In the second group do they perform the anastomosis underneath the pec major or they spread the muscle fibers?

Response: Thank you for the comments. In both groups, the anastomoses were performed either in the third or the fourth intercostal space. In Group 2, the anastomoses were performed beneath the pectoralis major muscle flap. These were added to the manuscript.

2) When rolling the pec major do they suture the muscle to itself to keep the shape?

Response: Yes, sutures were added to the muscle flap itself to retain its shape. This was also added to the manuscript.

We hope these modifications will obtain your approval. Thank you very much for your consideration of this paper.

Yours sincerely,

Hidehiko Yoshimatsu

March 18, 2022

Reviewer 2 Report

I congratulate the authors for their work. A new surgical method is presented that completes the PAP flap use in breast reconstruction. The patients seem to appreciate, subjectively, the use of the pectoralis muscle flap to augment the neo-breast upper pole.  I look forward to the three-years final results. 

Author Response

Thank you for your constructive comments concerning our manuscript “ Filling the Upper Pole with the Pectoralis Major Muscle Flap in Profunda Femoris Artery Perforator Flap Breast Reconstruction. ” We have studied your comments carefully and made corrections, which we hope will meet your approval. We answered your questions or comments in detail in the following paragraphs.   Reviewer comment: I congratulate the authors for their work. A new surgical method is presented that completes the PAP flap use in breast reconstruction. The patients seem to appreciate, subjectively, the use of the pectoralis muscle flap to augment the neo-breast upper pole.  I look forward to the three-years final results.    Response: Thank you for the constructive comment. We are conducting a study with larger number of patients and longer follow-ups.  

We hope these modifications will obtain your approval. Thank you very much for your consideration of this paper.

Yours sincerely,

Hidehiko Yoshimatsu

March 18, 2022